# The Role of Immune Semaphorins in Sepsis—A Prospective Cohort Study

**DOI:** 10.3390/microorganisms12122563

**Published:** 2024-12-12

**Authors:** Branimir Gjurasin, Lara Samadan Markovic, Leona Radmanic Matotek, Neven Papic

**Affiliations:** 1Department for Intensive Care, University Hospital for Infectious Diseases “Dr. Fran Mihaljević”, 10000 Zagreb, Croatia; bgjurasin@bfm.hr; 2Department for Infectious Diseases, School of Medicine, University of Zagreb, 10000 Zagreb, Croatia; samadanlara@gmail.com; 3Department for Immunological and Molecular Diagnostics, University Hospital for Infectious Diseases “Dr. Fran Mihaljević”, 10000 Zagreb, Croatia; lradmanic@bfm.hr; 4Department for Viral Hepatitis, University Hospital for Infectious Diseases “Dr. Fran Mihaljević”, 10000 Zagreb, Croatia

**Keywords:** sepsis, semaphorin, SEMA3A, SEMA3C, SEMA3F, SEMA4D, SEMA7A, biomarkers, bacteriaemia, septic shock, immune response

## Abstract

In sepsis, a balanced pro-inflammatory and anti-inflammatory response results in the bacterial clearance and resolution of inflammation, promoting clinical recovery and survival. Semaphorins, a large family of secreted and membrane-bound glycoproteins, are newly recognized biomarkers and therapeutic targets in immunological and neoplastic disorders. Although semaphorins might also be a crucial part of host defense responses to infection, their role in sepsis is yet to be determined. This study aimed to analyze the association of serum semaphorin concentrations with sepsis severity and outcomes. Serum semaphorin concentrations (SEMA3A, SEMA3C, SEMA3F, SEMA4D, and SEMA7A) were measured in 115 adult patients with community-acquired sepsis and 50 healthy controls. While SEMA3A was decreased, SEMA3C, SEMA3F, SEMA4D, and SEMA7A were increased in sepsis patients. All analyzed SEMA showed good accuracy in identifying patients with sepsis. SEMA kinetics were related to sepsis complications; SEMA3A, SEMA3C, SEMA3F, and SEMA4D with respiratory failure; SEMA3C and SEMA7A with acute kidney injury; and SEMA3C and SEMA3F were related to septic shock. Importantly, SEMA3A, SEMA3C, SEMA4D, and SEMA7A were associated with 28-day mortality. In conclusion, we provide evidence that semaphorins are associated with sepsis course and outcomes.

## 1. Introduction

Sepsis is defined as a dysregulated host immune response to infection resulting in life-threatening organ dysfunction, often with significant long-term clinical sequelae [1]. This relatively broad definition of sepsis can make timely diagnosis difficult, typically requiring a constellation of clinical, laboratory, radiologic, and microbiologic data, often resulting in its late identification (or misdiagnosis), which is the most important contributor to unfavorable patient outcomes [2]. While initially thought of as a predominantly hyperinflammatory response to infections, it is becoming evident that sepsis is characterized by heterogeneous immune responses; many patients have a predominantly counterbalancing anti-inflammatory response leading to immunosuppression [3]. This complex pathophysiology is characterized by significant biological and clinical heterogeneity, with molecular regulators of host immune responses that are not completely understood. Therefore, there is an ongoing quest for the identification of sepsis subphenotypes with distinct immune profiles and ideal biomarkers of sepsis, which might be pivotal for enhancing patient outcomes by developing new treatment options targeting host immune response [1,4,5,6].

Semaphorins (SEMA) are secreted and membrane-bound signaling glycoproteins with a variety of functions in many organ systems [7,8]. They are defined by the presence of an extracellular Sema domain and are subdivided into eight classes, of which classes 3–7 are found in humans, and classes 1 and 2 are found only in invertebrates [7,8]. While initially described as regulators of neuronal development (named after the discovery of their axon guidance function), during the last two decades, their relevance in angiogenesis, cardiogenesis, and oncogenesis has become apparent [7]. Furthermore, their role as crucial modulators of immune responses is emerging, describing their crucial role in regulating T- and B-cell activation, promotion or inhibition of pro-inflammatory cytokine production, and regulation of cell migration and adhesion, among others [9,10]. This has already been investigated in the diagnostics and therapy of immunologic diseases and cancer [11,12]. Meanwhile, only a few studies examined their role in human infections, as in COVID-19 [13], chronic hepatitis C [14], and pneumonia [15]. Data from animal models suggested that class 3 semaphorins might be involved in regulating inflammation during a septic response through Toll-like receptor pathway activation [8], and class 7 semaphorins might coordinate neutrophil responses by promoting adhesion, chemotaxis, and transmigration [16].

Given their critical role in regulating immune responses, we hypothesize that immune semaphorins might be novel regulators and biomarkers of sepsis severity and mortality, which have not been explored before. Therefore, we performed a prospective cohort study that analyzed semaphorin kinetics in patients with community-acquired bacterial sepsis and correlated it with sepsis severity, complications, and outcomes.

## 2. Materials and Methods

### 2.1. Study Design, Patients, and Samples

This study was part of a prospective, observational, non-interventional cohort study (Project SepsisFAT, Clinical Study Identifier NCT06021743) that included adult patients with severe community-acquired bacterial infections who were consecutively hospitalized between January 2022 and May 2023 at the University Hospital for Infectious Diseases Zagreb (UHID) [17]. Patients with clinically suspected community-acquired bacterial sepsis who met the sepsis-3 criteria, i.e., organ dysfunction detected by a change in the total sequential organ failure assessment (SOFA) score ≥ 2 points, were included [1]. Patients under the age of 18, immunosuppressed, neoplastic diseases, active autoimmune diseases, HIV infection, chronic liver diseases, alcohol consumption defined as a daily intake ≥ 20 g daily for women and ≥30 g for men [18], COVID-19 within the previous 3 months (including COVID-19 during the hospital stay), and pregnancy were excluded.

Patients who entered the study had their serum sampled on the first and fifth days of hospitalization for SEMA concentration analysis. The samples were divided into aliquots to avoid repeated freeze/thaw cycles and stored at −80 °C until testing.

We also included 50 healthcare workers without symptoms of acute infectious disease, immunocompromising conditions, active autoimmune diseases or active malignancy, obesity, or history of alcohol abuse, and with negative HIV and viral hepatitis markers as healthy controls (HC).

This study conformed to the ethical guidelines of the Declaration of Helsinki and was approved by the School of Medicine, University of Zagreb Ethics Committee (code 641-1/19-02/01). All participants gave written informed consent.

### 2.2. Data Collection, and Laboratory and Clinical Data

At admission, demographic and comorbidity data (including cardiovascular, pulmonary, kidney, and neurological conditions), as well as a baseline clinical status, were collected. The results of the routine laboratory testing were analyzed as follows: C-reactive protein (CRP), procalcitonin, lactate, bilirubin, aspartate aminotransferase (AST), alanine aminotransferase (ALT), gamma-glutamyl transferase (GGT), alkaline phosphatase (ALP), lactate dehydrogenase (LDH), serum albumin concentration, white blood cell count (WBC), neutrophil-to-lymphocyte ratio, hemoglobin, blood urea nitrogen (BUN), serum creatinine, platelet count, glucose, prothrombin time/INR, fibrinogen, and urine analysis. Microbiological data included cultures or the molecular detection results of urine, blood, cerebrospinal fluid, and other samples, as well as antimicrobial susceptibility data. The Acute Physiology and Chronic Health Evaluation (APACHE) II and SOFA scores were calculated. The source of the sepsis was defined as the presumed primary location of infection according to clinical and laboratory signs, and accordingly, patients were classified into 5 groups as follows: bacterial pneumonia, skin and soft tissue infection, gastrointestinal tract infection, urinary tract, or unknown source. Patients with a CNS infection per study protocol were not included. As this was a non-interventional study, the patients were treated according to the standard of care and at the discretion of the supervising physician. The cohort was followed until discharge, and there was no long-term follow-up.

The clinical outcomes evaluated were in-hospital mortality, according to the differences in SEMA concentration kinetics, as well as septic shock, the degree of organ dysfunction, ICU admission, respiratory supports (mechanical ventilation), renal replacement therapy (dialysis), duration of hospitalization, and development of nosocomial infections > 48 h after hospital admission. These were correlated with the SEMA serum concentrations.

### 2.3. Measurement of Semaphorin Serum Concentrations

Semaphorins were quantified using a standardized enzyme-linked immunosorbent assay (ELISA) (Human Semaphorin-3A, -3C, -3F, -4D and -7A by ELISA kit, AssayGenie, Dublin, Ireland, and Human Semaphorin-3C ELISA Kit, MyBioSource, San Diego, CA, USA), following the manufacturer’s recommendations.

### 2.4. Statistical Analysis

Data were analyzed and presented descriptively as medians with interquartile ranges (IQR) or frequencies. Fisher’s exact test and the Mann–Whitney U test were used to compare the two groups. The repeated measures two-way ANOVA test with Tukey’s multiple comparisons test was used to analyze the kinetics of the SEMA between two groups at 2 time points (day 1 and day 5). The sample size of a minimum of 100 participants was selected according to power analysis for the RM-ANOVA test to achieve an 80% chance of detecting a difference in semaphorin concentrations at a 5% significance level. A Wilcoxon matched pairs test was used to analyze the changes in concentrations at the baseline and on day 5. Correlations were analyzed using Spearman’s rank correlation coefficient and summarized in a correlation matrix. Survival analysis was evaluated using the Kaplan–Meier method, and a comparison between groups was made using the log-rank test. Statistical analyses were performed using GraphPad Prism Software version 10 (San Diego, CA, USA).

## 3. Results

### 3.1. Baseline Patients’ Characteristics

Overall, 115 patients with sepsis (53 males (46%); median age 64, IQR 53–74 years) and 50 controls (25 males (50%); median age 59, IQR 48–65 years) were included, as presented in Table 1. The most common comorbidities in the patients with sepsis were arterial hypertension (64, 55.7%), T2DM (29, 25.2%), dyslipidemia (24, 20.9%), and visceral obesity as measured using a waist-to-height ratio (78, 67.8%) (Table 1). A significant number of patients fulfilled the criteria for metabolic syndrome (with a median of 3, IQR 2–4 number of metabolic syndrome components present per patient).

The time from onset of symptoms to hospital admission was 4 (2–6) days, with moderate disease severity, as measured by the SOFA (3, IQR 2–6) and APACHE II scores (14, IQR 8–22). The most common source of sepsis was the lower respiratory tract (40, 34.8%), followed by skin (22, 19.1%), and urinary tract infections (20, 17.4%). Bacteriaemia was present in 51 (44.3%) patients, with the most common isolate being *Escherichia coli* (14, 12.2%), followed by *Staphylococcus aureus* (12, 10.4%) and *Streptococcus pneumoniae* (9, 7.8%) (Table 1). In addition, urine cultures were positive in 25 patients, respiratory tract cultures in 14 patients, stool cultures in 6, and other samples (swabs, aspirates, or punctate) in 4 patients. Overall, combining all available microbiological results, bacterial etiology was confirmed in 66 (57.4%) patients.

As shown in Table 2, patients had significantly increased inflammatory markers such as CRP (230 mg/L, IQR 154–304), procalcitonin (1.7 µg/L, IQR 0.28–8.97), IL-6 (239.47 pg/mL, IQR 97.99–859.22), and fibrinogen (6.1 g/L, IQR 5.1–8.17), as well as leukocytosis and neutrophilia (WBC of 14.1 × 10^9^/L, IQR 10.1–18.45 and neutrophils of 12.2 × 10^9^/L, IQR 8.4–15.6). A significant proportion of patients had signs of kidney injury, as measured by the eGFR with a median of 52.7 (IQR 28.8–75.2) mL/min/1.73 m^2^ (44.3% of patients had eGFR < 50 mL/min/1.73 m^2^). An AST above the reference ranges was in 54 (46.9%), ALT in 47 (40.9%), and GGT in 56 (48.7%) patients. Coagulopathy, as defined by an INR > 1.3, was detected in 16 (13.9%) patients.

### 3.2. Analysis of Serum Semaphorin Concentrations in Patients with Sepsis and Healthy Controls

Serum concentrations of semaphorins SEMA3A, SEMA3C, SEMA3F, SEMA4D, and SEMA7A were detectable in all patients (Figure 1). Patients with sepsis had significantly higher serum levels of SEMA3C, SEMA3F, SEMA4D, and SEMA7A and a significantly lower SEMA3A (Figure 1a, Table 3).

A ROC analysis was performed to determine the cutoff values of the serum semaphorin concentrations at admission for differentiating patients with sepsis from the healthy controls (Figure 1b and Table 4). SEMA3F and SEMA7A showed good accuracy in distinguishing patients with sepsis from the healthy controls. A cutoff value of SEMA3F > 4.2 ng/mL correctly predicted sepsis with a sensitivity of 87% and specificity of 90% (AUC 0.90, 95%CI 0.84–0.95). SEMA7A > 1.3 ng/mL predicted sepsis with a sensitivity of 80% and specificity of 90% (AUC 0.92, 95%CI 0.87–0.96), and SEMA3C > 1.25 ng/mL had a sensitivity of 81% and specificity of 73% (AUC 0.83, 95%CI 0.76–0.90). SEMA3A (AUC 0.68, 95%CI 0.60–0.76) and SEMA4D (AUC 0.71, 95%CI 0.60–0.82) had lower AUC scores (Table 4).

Next, we examined the differences in semaphorin concentrations at admission regarding the source of sepsis. Septic patients were stratified according to the source of sepsis, as presented in Figure 2a. The sepsis source did not have a significant impact on the serum semaphorin concentrations.

Similarly, in a subgroup of patients with positive blood cultures, there were no differences in SEMA concentrations based on the pathogen detected (Figure 2b).

### 3.3. Correlation Analysis of Baseline Serum Semaphorin Concentrations with Routine Clinical and Laboratory Parameters

We analyzed the potential correlations among paired laboratory parameters, including the semaphorin concentrations and clinical variables in patients with sepsis, as presented in Figure 3.

Serum SEMA3A negatively correlated with SEMA7A (*r* = −0.30, *p* < 0.01) and positively with SEMA4D (*r* = 0.27, *p* < 0.01). SEMA3F positively correlated with SEMA4D (*r* = 0.38, *p* < 0.01), while SEMA4D negatively correlated with SSEMA7A (*r* = −0.28, *p* < 0.01). Regarding the sepsis severity scores, SEMA3C correlated with SOFA (*r* = 0.34, *p* < 0.001), while SEMA4D (*r* = −0.2, *p* = 0.04) and SEMA7A (*r* = 0.17, *p* = 0.05) correlated with the number of systemic inflammatory response syndrome criteria (SIRS score [19]). No analyzed semaphorin correlated with CRP or IL-6, but SEMA3C correlated positively with PCT (*r* = 0.23, *p* = 0.01) and lactate (*r* = 0.19, *p* = 0.05), and SEMA3F (*r* = 0.17, *p* = 0.04) and SEMA7A with fibrinogen (*r* = 0.17, *p* = 0.04). SEMA3C negatively correlated with the platelet count (*r* = −0.23, *p* = 0.01), SEMA4D with the WBC (*r* = −0.25, *p* = 0.01) and neutrophil count (*r* = −0.22, *p* = 0.03), and SEMA7A correlated positively with the lymphocyte count (*r* = 0.17, *p* = 0.04). SEMA3C (*r* = −0.25, *p* < 0.01) and SEMA7A (*r* = −0.16, *p* = 0.04) negatively correlated with eGFR. Regarding liver enzymes, the AST showed a correlation with SEMA4D (*r* = 0.20, *p* = 0.04), ALT with SEMA3F (*r* = 0.17, *p* = 0.03), and GGT with SEMA3A (*r* = −0.16, *p* = 0.04) and SEMA3F (*r* = 0.32, *p* < 0.01).

### 3.4. Changes in Serum Semaphorin Concentration on the Fifth Day of Hospitalization

Further, we analyzed the concentrations of serum SEMA on day 5 of hospitalization in paired patient samples. As shown in Figure 4, SEMA3A (median of difference 1.9, 95%CI 0.2–3.7), SEMA3C (median of difference 0.2 95%CI 0.05–0.4), and SEMA3F (median of difference 0.5, 95%CI −0.07–1.2) significantly decreased in the paired samples, while there were no significant differences in the SEMA4D and SEMA7A concentrations.

In correlation analysis, SEMA3C correlated with the SOFA score (*r* = 0.31, *p* < 0.01); SEMA3C (*r* = 0.18, *p* < 0.01), SEMA4D (*r* = −0.22, *p* = 0.03), and SEMA7A (*r* = 0.19, *p* = 0.02) with SIRS; and SEMA3C (*r* = 0.18, *p* = 0.04) and SEMA7A (*r* = 0.16, *p* = 0.05) with the APACHE II score. SEMA3C correlated with the WBC (*r* = 0.36, *p* < 0.01), neutrophil (*r* = 0.26, *p* < 0.01), and lymphocyte (*r* = −0.19, *p* = 0.04). SEMA3C (*r* = −0.23, *p* = 0.02) and SEMA7A (*r* = 0.28, *p* < 0.01) correlated with the platelet count, as well as with eGFR (*r* = 0.26, *p* = 0.01).

### 3.5. Association of Serum Semaphorin Kinetics with Sepsis Severity, Complications, and Outcomes

We further examined the impact of the baseline and day 5 serum semaphorin concentrations on sepsis complications. On day 5 of hospitalization, 108 patients remained hospitalized (5 patients died and 2 were discharged), and the semaphorin concentrations were measured in paired patients’ sera. A series of RM two-way ANOVA analyses were performed to identify the differences in the semaphorin kinetics associated with sepsis complications, as shown in Figure 5.

Seventeen patients had shock, requiring vasopressor therapy for longer than 24 h (median of 5, IQR 1–8 days). SEMA3C was significantly higher on both days 1 and 5 in patients with shock. Patients with shock had a decrease in SEMA3F on day 5, while patients without shock did not.

During hospitalization, 32 patients developed moderate/severe ARDS and required mechanical ventilation for a median duration of 8 days (IQR 5–16 days). As shown in Figure 5, the patients with ARDS had significantly lower serum concentrations of SEMA3A and higher SEMA3C on both days 1 and 5 of hospitalization. SEMA4D was significantly higher in the non-ARDS group on day 5. Patients who developed ARDS had a significant decrease in SEMA3F on the fifth day, while patients in the non-ARDS group had a significant decrease in SEMA3A.

Thirty-four patients developed stage 3 acute kidney injury (AKI), and fifteen of them required renal replacement therapy (RRT) for a median duration of 4 days (3–9 days). SEMA3C was significantly higher on days 1 and 5, and SEMA7A was lower on day 1 in patients with AKI. In patients without AKI, SEMA3A significantly decreased on day 5, in contrast to the patients without AKI.

Next, we analyzed the serum semaphorin concentrations with the development of nosocomial infections as a complication of sepsis. A total of 22 patients developed nosocomial infections during hospitalization (*Cl. difficile* infection in 6, VAP in 7, UTI in 10, and catheter-associated bacteriaemia in 5). In patients with nosocomial infections, SEMA3C was significantly higher, whereas SEMA3A and SEMA4D were lower on day 5.

### 3.6. Association of Serum Semaphorin Concentration Kinetics with Sepsis Mortality

We examined the impact of the baseline and day 5 serum semaphorin concentrations on in-hospital mortality. Overall, 22 (19.1%) patients in our cohort died during hospitalization; the 7-day mortality was 5.2% (*n* = 6 patients), 14-day 10.4% (*n* = 12), and 28-day 13.9% (*n* = 16).

On admission, the non-survivors had a significantly lower SEMA3A (8.3, IQR 4.0–16 vs. 13, IQR 7.6–20), while SEMA3C (2.0, IQR 1.8–2.3 vs. 1.6, IQR 1.3–1.9) and SEMA7A (2.4, IQR 2.0–3.7 vs. 1.8, IQR 1.3–2.7) were higher. SEMA4D was significantly higher on day 5 in survivors (3.5, IQR 2.5–4.9 vs. 5.0, IQR 4.0–6.0), as shown in Figure 6.

The predictive capability of the inflammatory markers and scoring systems for the 28-day mortality risk was assessed using ROC analysis, and their cutoff values with specificity and sensitivity are presented in Figure 7.

The serum concentrations of SEMA3C on day 1 (AUC 0.75, 95%CI 0.63–0.87, *p* = 0.0025) and day 5 (AUC 0.72, 95%CI 0.58–0.86, *p* = 0.0068) and SEMA7A on day 1 (AUC 0.72, 95%CI 0.60–0.84, *p* = 0.0051) showed good discriminatory values in predicting mortality. SOFA had an AUC of 80.3 (95%CI 0.70–0.90, *p* = 0.001) and APACHE II had 0.85 (95%CI 0.76–0.93). Regarding the routinely measured inflammatory markers, only PCT had an AUC > 0.6 (AUC 0.75, 95%CI 0.62–0.89).

Next, in survival analysis using Kaplan–Meier estimates, a SEMA3C > 1.8 ng/mL on day 5 (HR 5.3, 95%CI 1.81–15.64, *p* = 0.0004, Figure 8a) and a SEMA7A concentration > 2.2. (HR 5.19, 95%CI 1.88–14.36, *p* = 0.0012, Figure 8b) on day 1 appeared to be efficient prognostic markers associated with 28-day mortality.

## 4. Discussion

Here, we provide novel insights into the role of semaphorins on the course and outcomes of sepsis. Patients with sepsis had different SEMA serum concentrations as compared to the healthy controls, and those might be related to clinically relevant outcomes. While SEMA3A was decreased, SEMA3C, SEMA3F, SEMA4D, and SEMA7A were increased in sepsis patients, and all analyzed SEMA showed good accuracy in identifying patients with sepsis. Furthermore, the kinetics of SEMA were related to sepsis complications; SEMA3A, SEMA3C, SEMA3F, and SEMA4D were related to respiratory failure; SEMA3C and SEMA7A were related to acute kidney injury, while SEMA3C and SEMA3F were related to septic shock. Importantly, SEMA3A, SEMA3C, SEMA4D, and SEMA7A were associated with 28-day mortality.

Surprisingly, although the association of SEMA with different immunological conditions has been extensively studied, to date, there are only a few reports on SEMA and human infections [13,14,16]. Semaphorins were reported as potential biomarkers for the severity of chronic hepatitis C infection, and SEMA3A and SEMA6D correlated with the fibrosis stage [14]. The association of SEMA was also analyzed in patients with COVID-19. The authors reported higher serum concentrations of SEMA3C, SEMA3F, and SEMA7A and lower serum concentrations of SEMA3A, which correlated with disease severity [13]. A more recent study measured the SEMA7A levels in 14 patients with ARDS and suggested an association of SEMA7A with oxygen levels and organ dysfunction scores [16]. Here, we provide the first clinical data on SEMA concentrations in a well-defined cohort of patients with community-acquired sepsis. There are several possible pathophysiological explanations for our findings.

The role of SEMA3A in inflammation was explored in several mice models, however, with conflicting results. While some studies showed immunoinhibitory functions of SEMA3A being associated with reduced inflammation [20] and a milder lung injury in ARDS [21], in other models, SEMA3A was associated with an infiltration of neutrophils and macrophages in kidney tissue [22,23]. Blocking SEMA3A in sepsis led to a decreased secretion of TNF-α and IL-6, which was associated with lower mice mortality [24,25]. In our cohort, the lowest SEMA3A levels were observed in the most severely ill patients, which is in line with their immunosuppressive functions reported in some studies, such as inhibition of Th1 lymphocyte activation and neutrophil migration and the stimulation of macrophage transformation from a classically activated (M1) to resolution phenotype [26].

Other class 3 semaphorins, such as SEMA3C, were increased in patients with sepsis, correlated with the WBC, neutrophil, and lymphocyte counts, and showed different serum kinetics in survivors who had a significant decrease in serum concentrations on day 5, in contrast to non-survivors. Except for a few reports on its role in dendritic cell migration, angiogenesis, and oncogenesis [12], there are no data on its function in inflammation. Some authors suggest that SEMA3C might have similar inflammatory functions as SEMA3F, for which we have more reports. SEMA3F promotes neutrophil migration and pro-inflammatory cytokine secretion [27]. In animal models, exogenous intratracheal administration of lipopolysaccharide (LPS) leads to increased concentrations of neutrophils and SEMA3F in bronchoalveolar fluid [27]. On the other hand, a neutrophil-specific loss of SEMA3F results in more rapid neutrophil recruitment and clearance from the lungs [27]. Similarly, we observed an association of SEMA3F with lung dysfunction and increased SEMA3F levels in patients with ARDS and shock.

Next, we found increased SEMA4D levels in patients with sepsis that failed to increase further on day 5 in patients who developed ARDS and in non-survivors. SEMA4D was the first semaphorin described to have an immunoregulatory function [28], as extensively reviewed in [8]. Sema4D acts as a negative regulator of neutrophil activation, fine-tunes T- and B-lymphocyte responses, stimulates the differentiation of macrophages and dendritic cells, and, in some contexts, pro-inflammatory cytokine secretion [8]. Here, we provide additional clues on its role in regulating complex inflammatory networks in sepsis that warrant further studies.

The class 7 semaphorin, SEMA7A, stimulates Th1 and Th17 lymphocyte differentiation, neutrophil migration, and pro-inflammatory cytokine secretion [8]. Animal data have shown that SEMA7A stimulates transendothelial neutrophil migration in the lung and peritoneal tissue, as well as reversal of this process by blocking SEMA7A [15,29,30]. A recent study in animal models described an association between SEMA7A and oxygenation levels in ARDS, as well as an induction of the neutrophil chemotaxis effect of SEMA7A during pneumonia [16]. Similarly, in our cohort, SEMA7A was associated with sepsis severity, adverse outcomes, and mortality.

Our study should be viewed within its limitations. Given its observational design, causation could not be established; sepsis should be viewed as a heterogeneous syndrome with multiple factors influencing its outcomes. Therefore, a relatively small number of participants in our cohort limits the statistical analysis and should be confirmed in a larger population. Since a majority of patients with sepsis progress to multiorgan failure, the association of semaphorins with specific organ damage should be viewed within this context. The impact of antibiotic selection or ICU treatment regiments on semaphorin kinetics was not analyzed; concentrations of semaphorins were determined on days 1 and 5, which might not reflect the changes associated with the resolution phase of sepsis or late complications. An association of SEMA with sepsis subphenotypes and different etiologies due to a low number of blood-culture positivity was not assessed. Furthermore, serum semaphorin concentrations in sepsis could be influenced by patients’ chronic conditions, such as allergic or rheumatologic diseases, components of metabolic syndrome, or neoplastic diseases, which were not assessed here. Regardless, we report novel data on the semaphorins’ profile in patients with sepsis, which should be confirmed in larger multicentric studies aimed at better understanding the role of SEMA in serious infections.

## 5. Conclusions

In conclusion, we have shown that patients with sepsis have different serum concentrations of SEMA3A, SEMA3C, SEMA3F, SEMA4D, and SEMA7A compared to healthy controls and that their kinetics during sepsis correlate with disease severity and outcomes. This justifies further investigations on semaphorins as new diagnostic and prognostic biomarkers and potential therapeutic targets of sepsis.

## Figures and Tables

**Figure 1 microorganisms-12-02563-f001:**
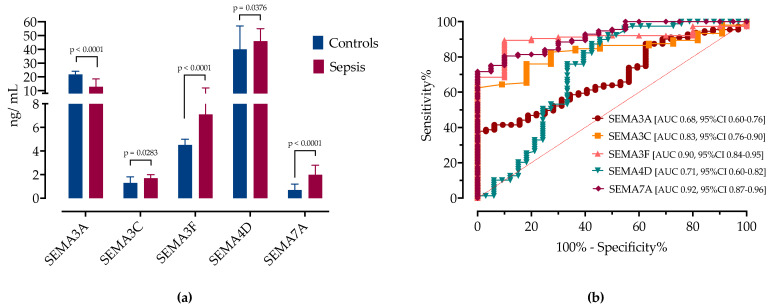
(**a**) Serum concentrations of semaphorin in healthy controls and patients with sepsis at hospital admission. (**b**) ROC curve analysis of serum semaphorins for determination of sepsis. AUCs are shown with corresponding 95% CIs.

**Figure 2 microorganisms-12-02563-f002:**
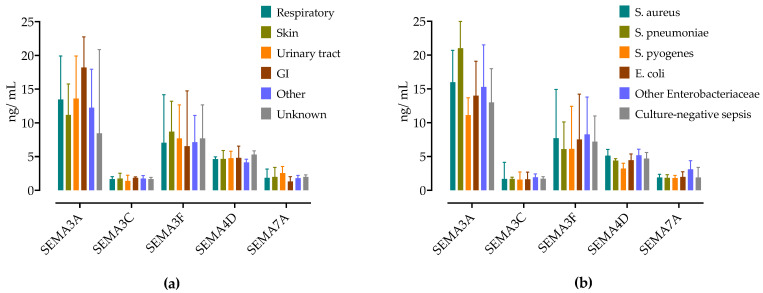
(**a**) Serum concentrations of semaphorin in patients with sepsis stratified by infection source and (**b**) by detected bacteria in blood cultures. Medians are shown with IQRs.

**Figure 3 microorganisms-12-02563-f003:**
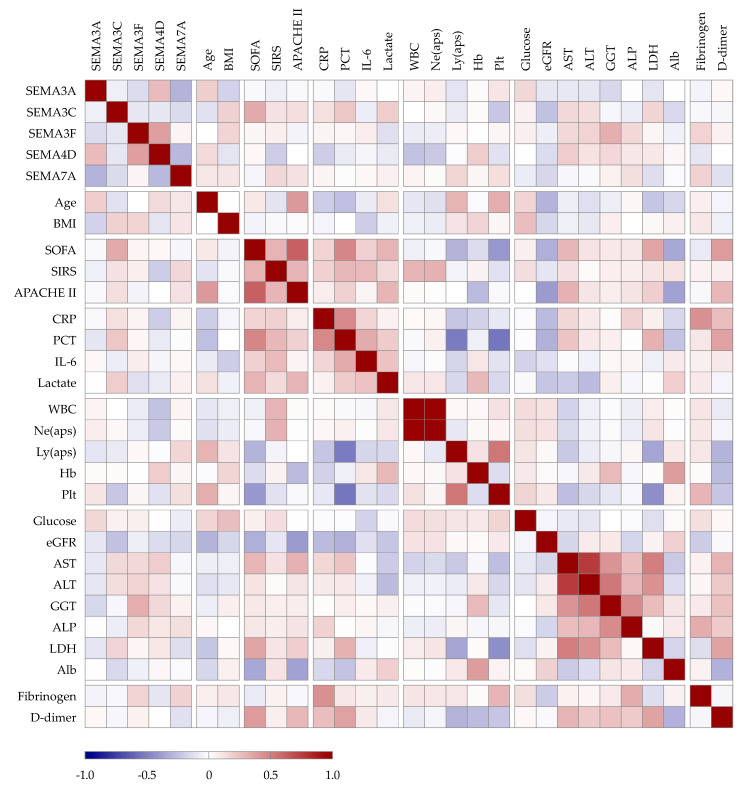
Spearman correlation heatmap. The strength of the correlation between two variables is represented by the color at the intersection of those variables. Colors range from dark blue (strong negative correlation; *r* = −1.0) to red (strong positive correlation; *r* = 1.0).

**Figure 4 microorganisms-12-02563-f004:**
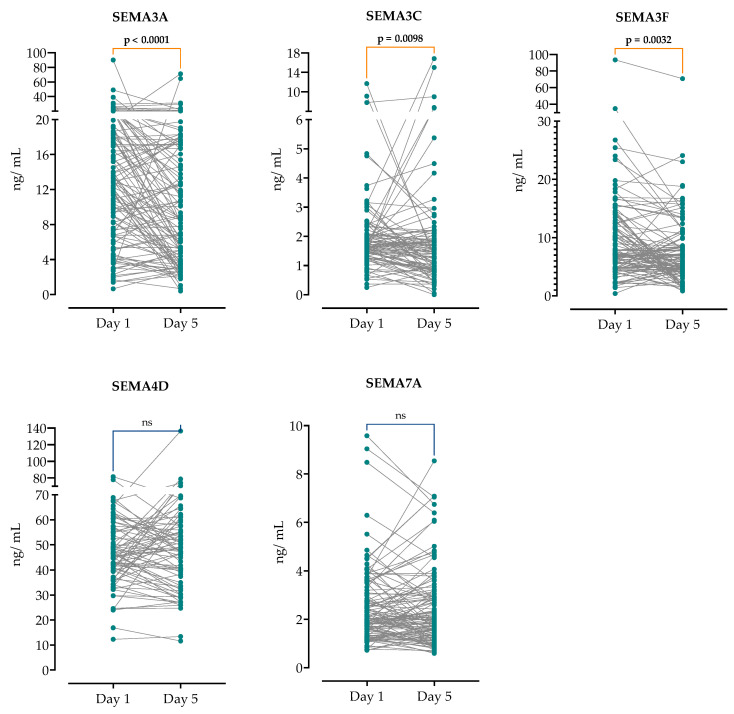
Comparison of semaphorin concentrations on admission and at day 5. Wilcoxon rank sum test was performed to analyze the difference in time within the groups. Note: ns, non-significant difference (*p* > 0.05).

**Figure 5 microorganisms-12-02563-f005:**
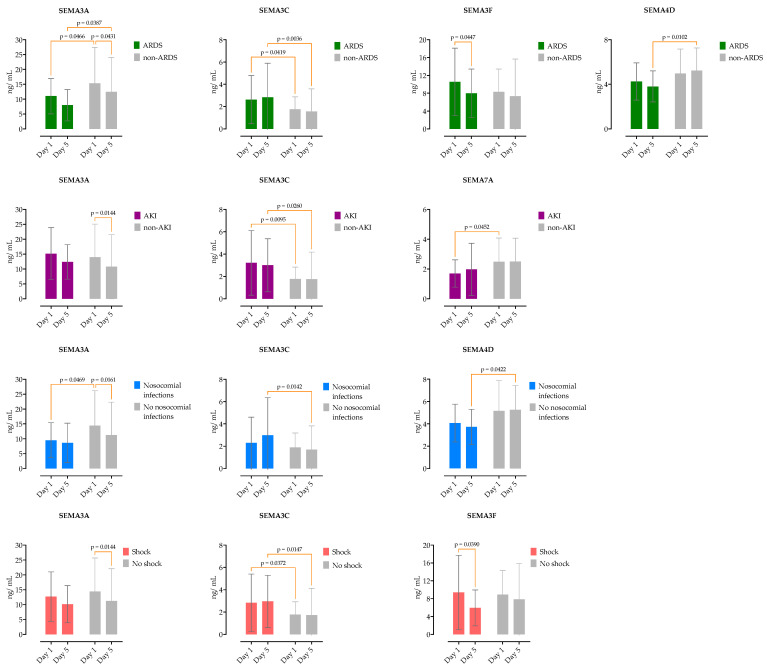
Comparison of semaphorin serum concentrations at two selected time points (day 1 and day 5), stratified by the presence of sepsis complications. Medians are shown with IQRs. Repeated measured two-way ANOVA with Tukey’s multiple comparisons test was used to calculate the source of variations.

**Figure 6 microorganisms-12-02563-f006:**
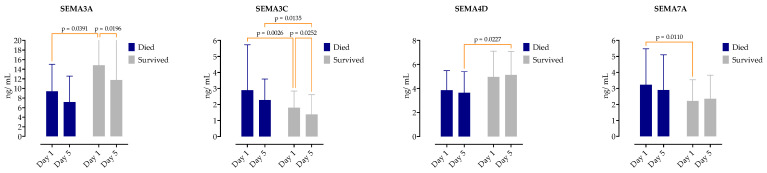
Comparison of semaphorin serum concentrations at two selected time points (day 1 and day 5) between survivors and non-survivors. Medians are shown with IQRs. Repeated measured two-way ANOVA with Tukey’s multiple comparisons test was used to calculate the source of variations.

**Figure 7 microorganisms-12-02563-f007:**
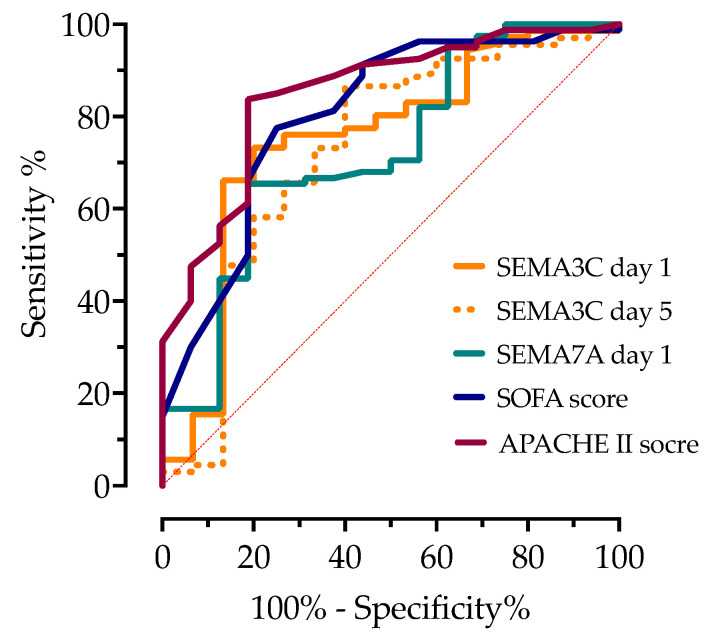
ROC curve analysis of serum semaphorin concentrations and sepsis severity scores for predicting 28-day mortality. AUC is shown with the corresponding 95%CI.

**Figure 8 microorganisms-12-02563-f008:**
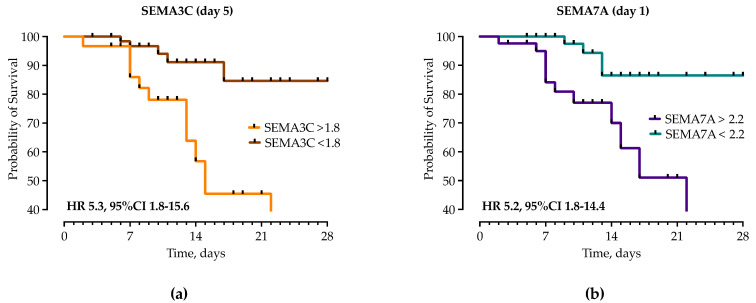
Kaplan–Meier curves and hazard ratios (HR) with corresponding 95% confidence intervals (95% CI) for the probability of 28-day survival stratified by the presence of (**a**) SEMA3C and (**b**) SEMA7A and showing significant associations with semaphorin serum concentrations and survival.

**Table 1 microorganisms-12-02563-t001:** Baseline patients’ characteristics.

	Patients with Sepsis (*n* = 115) *
Male sex	53 (46.1%)
Age, years	64 (53.5–74)
BMI, kg/m^2^	27.7 (24.6–34.5)
Smoking	24 (20.9%)
Comorbidities	
Charlson comorbidity index	3 (1–5)
Diabetes mellitus	29 (25.2%)
Arterial hypertension	64 (55.7%)
COPD	6 (5.2%)
Dyslipidemia	24 (20.9%)
Gastritis/GERD	9 (7.8%)
Cardiovascular diseases	32 (27.8%)
Chronic renal disease	8 (7.0%)
Neurological disease	14 (12.2%)
Disease severity at admission	
Duration of illness, days	4 (2–6)
SOFA	3 (2–6)
SIRS	3 (2–3)
APACHE II score	14 (8–22)
Infection source	
Pneumonia	40 (34.8%)
Skin and soft tissue	22 (19.1%)
Gastrointestinal tract	14 (12.2%)
Urinary tract	20 (17.4%)
Other	9 (7.8%)
Unknown	10 (8.7%)
Bacteriaemia	51 (44.3%)
Isolates in the blood cultures	
*S. aureus*	12 (10.4%)
*S. pneumoniae*	9 (7.8%)
*Streptococcus pyogenes*	5 (4.3%)
*Enterococcus* spp.	3 (2.6%)
*E. coli*	14 (12.2%)
*Klebsiella pneumoniae*	5 (4.3%)
Other Enterobacteriaceae	3 (2.6%)

* Data are presented as frequencies (%) or medians with IQRs.

**Table 2 microorganisms-12-02563-t002:** Baseline laboratory findings.

	Patients with Sepsis *
CRP, mg/L	230 (154–304)
Procalcitonin, µg/L	1.7 (0.28–8.97)
Interleukin-6, pg/mL	239.47 (97.99–859.22)
Lactate, mmol/L	1.97 (1.27–2.93)
WBC, ×10^9^/L	14.1 (10.1–18.45)
Lymphocyte count, 10^9^/L	0.83 (0.46–1.31)
Neutrophil count, 10^9^/L	12.19 (8.37–15.64)
Hemoglobin, g/L	123 (108–135)
Platelets, ×10^9^/L	216 (146–272)
Glucose, mmol/L	7.3 (6.2–9.7)
Urea, mmol/L	7.0 (5.0–12.5)
Creatinine, µmol/L	79 (63–148.5)
eGFR, mL/min/1.73 m^2^	52.7 (28.8–75.2)
Bilirubin, µmol/L	13 (9–19)
AST, IU/L	35 (22–73)
ALT, IU/L	30 (19–62)
GGT, IU/L	44 (24–106)
LDH, IU/L	225 (184–302)
Albumins, g/L	31.9 (27.5–36.9)
INR	1.04 (0.97–1.17)
Fibrinogen, g/L	6.1 (5.1–8.17)
D-dimer, mg/L	2.38 (1.07–4.24)

* Data are presented as medians with interquartile ranges (IQR). Abbreviations: C-reactive protein (CRP); white blood cell count (WBC); estimated glomerular filtration rate (eGFR); Aspartate aminotransferase (AST); Alanine aminotransferase (ALT); gamma-glutamyl transferase (GGT); international normalized ratio (INR); lactate dehydrogenase (LDH).

**Table 3 microorganisms-12-02563-t003:** Serum concentrations of semaphorins in healthy controls and patients with sepsis.

	Healthy Controls (*n* = 50)	Patients with Sepsis (*n* = 115)	Difference (95% CI)	*p*-Value
SEMA3A, ng/mL	21.8 (16.2–24.1)	12.8 (6.8–18.5)	−9.0 (−10–−6.1)	<0.0001
SEMA3C, ng/mL	1.3 (1.1–1.8)	1.7 (1.3–2.0)	0.4 (0.03–0.6)	0.0283
SEMA3F, ng/mL	4.5 (2.7–5.0)	7.1 (4.7–12)	2.6 (1.9–5.1)	<0.0001
SEMA4D, ng/mL	40 (28–57)	46 (39–55)	6.3 (0.3–12)	0.0376
SEMA7A, ng/mL	0.69 (0.27–1.2)	2.0 (1.4–2.8)	1.3 (1.0–1.6)	<0.0001

**Table 4 microorganisms-12-02563-t004:** ROC analysis of sensitivity and specificity in differentiating sepsis patients from healthy controls.

	Sensitivity (95% CI)	Specificity (95% CI)	AUC (95% CI)	*p*-Value
SEMA3A < 14 ng/mL	58.5% (51.6–66.6%)	62.5% (49.8–75.0%)	0.6855 (0.60–0.76)	0.0014
SEMA3C > 1.25 ng/mL	80.7% (73.6–86.3%)	72.7% (47.9–88.5%)	0.8304 (0.76–0.90)	0.0003
SEMA3F > 4.2 ng/mL	86.8% (80.7–91.2%)	90.0% (65.2–98.9%)	0.9026 (0.84–0.95)	<0.0001
SEMA4D > 40 ng/mL	72.1% (63.2–79.6%)	66.7% (52.4–78.4%)	0.7127 (0.60–0.82)	0.0004
SEMA7A > 1.3 ng/mL	80.5% (73.7–85.9%)	90.0% (73.8–97.3%)	0.9177 (0.87–0.96)	<0.0001

## Data Availability

The raw data supporting the conclusions of this article will be made available by the authors on request. The data are not publicly available due to privacy or ethical restrictions.

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
