# Peer review of "The Role of Immune Semaphorins in Sepsis—A Prospective Cohort Study"

_microorganisms, 2024, doi:10.3390/microorganisms12122563_

Round 1

Reviewer 1 Report

Comments and Suggestions for Authors

Branimir Gjurasin and co-authors investigated on the association of serum semaphorin concentrations with sepsis severity and outcomes. Their study revealed that patients with sepsis have different serum concentrations of SEMA3A, SEMA3C, SEMA3F, SEMA4D and SEMA7A when compared to healthy controls, and that their kinetics during sepsis correlate with disease severity and outcomes. Specifically, While SEMA3A was decreased, SEMA3C, SEMA3F, SEMA4D and SEMA7A were increased in sepsis patients. In addition, kinetics were found to be related to sepsis complications.

The authors should be commended for their work and their findings, that add more information on the role of SEMA in the pathogenesis and evolution of sepsis. The overall quality of the manuscript is good, fluent to the readers. The statistical analysis seems appropriate for the purpose of this research. I would suggest the authors to add information on the microbiological tests results in a separate table. 

I do not have other suggestions. Only minor revision is required. 

Author Response

REVIEWER 1

Branimir Gjurasin and co-authors investigated on the association of serum semaphorin concentrations with sepsis severity and outcomes. Their study revealed that patients with sepsis have different serum concentrations of SEMA3A, SEMA3C, SEMA3F, SEMA4D and SEMA7A when compared to healthy controls, and that their kinetics during sepsis correlate with disease severity and outcomes. Specifically, While SEMA3A was decreased, SEMA3C, SEMA3F, SEMA4D and SEMA7A were increased in sepsis patients. In addition, kinetics were found to be related to sepsis complications.

The authors should be commended for their work and their findings, that add more information on the role of SEMA in the pathogenesis and evolution of sepsis. The overall quality of the manuscript is good, fluent to the readers. The statistical analysis seems appropriate for the purpose of this research. I would suggest the authors to add information on the microbiological tests results in a separate table. 

I do not have other suggestions. Only minor revision is required. 

Authors’ response: We thank the reviewer for positive comments. We agree that adding microbiological test results would improve the manuscript. In Table 1. blood cultures isolates are now added. In addition, microbiological results from other samples (urinary culture, respiratory samples, stool, swabs or punctate) are now provided in the Results.

Reviewer 2 Report

Comments and Suggestions for Authors

Microorganisms 3317812

Very interesting but not correctly presented.

Patients’ characteristics is not sufficiently described.  

How about treatment-related changes in semaphorins kinetics?

Relevance between sepsis complication and type of the semaphorins seems to be  within statistical error related to too low number of the participants. In most patients, sepsis progresses to multiorgan failure involving not one organ.

The value of the study was limited by the very low number of positive blood cultures (44%). It may suggest that sepsis was not confirmed and may explain “moderate disease severity”. Also, it is not specified as how the infection source was diagnosed.     

The authors are right that their study was based on too-low number of the participants and therefore it should be treated as a preliminary observation.   

Detailed:

-              Line 114: what the authors named “renal replacement therapy”? Renal dialysis?

-              Line 115: if the patients were admitted with sepsis signs and symptoms and their infection was classified as community-related, which infections were named “hospital-acquired infections” please explain. This group of the patients with superinfections is not described.   

-              Line 148: how many patients survived up 6th day of the study, and how many completed the study on day 28th? This information should be given here.    

-              Figure 2(b): why there is no information about culture-negative patients?

-              Line 367: lung injury or disfunction?

-              Line 392: the patients’ cohort was not well-defined and too many culture-negative cases were included to the study.          

Author Response

REVIEWER 2

Microorganisms 3317812

Very interesting but not correctly presented.

Patients’ characteristics is not sufficiently described.  

Authors’ response: We thank the reviewer for this comment. Patients’ characteristics are now described in more detail, including laboratory and microbiological data.  

How about treatment-related changes in semaphorins kinetics?

Authors’ response: We agree with the reviewer that biomarkers’ kinetics should ideally be explored within the context of different treatment regiments. However, our study was not designed to answer the question if different antibiotic selections or ICU treatments influence semaphorin kinetics (e.g. E.coli urosepsis treated with ceftriaxone or pip/tazo, or the effect of CRRT). Due to the limited number of patients, we are unable to perform this kind of analysis. Furthermore, a vast majority of our patients received appropriate empirical treatment, so we could not analyze the impact of inappropriate antibiotic selection on semaphorin kinetics. We agree that this should be further explored in larger follow-up studies. We have added this concerns in our limitations.  

Relevance between sepsis complication and type of the semaphorins seems to be within statistical error related to too low number of the participants. In most patients, sepsis progresses to multiorgan failure involving not one organ.

Authors’ response: We thank the reviewer for this comment. Sepsis progresses to multiorgan failure in most patients. Therefore, a specific correlation between semaphorin concentrations and specific organ damage is difficult to analyze and our data cannot provide a causal relationship, as already stated in limitations. Indeed, the specific semaphorin concentrations were associated with parameters of multiple organ damage (e.g. SEMA3A with shock, ARDS, AKI or nosocomial infection). This is now further highlighted in study limitations.     

The value of the study was limited by the very low number of positive blood cultures (44%). It may suggest that sepsis was not confirmed and may explain “moderate disease severity”. Also, it is not specified as how the infection source was diagnosed.     

Authors’ response: The blood culture positivity was not predefined inclusion criteria for our study. We defined sepsis according to Sepsis-3 criteria that do not require presence of bacteriaemia. The blood cultures positivity rate of 44% was expected and similarly reported in other studies examining community-acquired sepsis, where etiology is usually identified in less than 50% (Crit Care 2021;25:167 , Critical Care Medicine 2021;49(11):e1063-1143). As stated, our results might not reflect the semaphorins concentrations in other clinical settings, e.g. surgical, ICU, immunocompromised patients, where bacteriaemia might be more prevalent. The definition of sepsis source is now added in Methods.

Detailed:

-              Line 114: what the authors named “renal replacement therapy”? Renal dialysis? Authors’ response: Clarified.

-              Line 115: if the patients were admitted with sepsis signs and symptoms and their infection was classified as community-related, which infections were named “hospital-acquired infections” please explain. This group of the patients with superinfections is not described.

Authors’ response: Clarified.    

-              Line 148: how many patients survived up 6th day of the study, and how many completed the study on day 28th? This information should be given here.    

Authors’ response: We thank the reviewer for making this point. These data are now added in the results section (3.5).      

-              Figure 2(b): why there is no information about culture-negative patients?

Authors’ response: the semaphorin concentrations in culture-negative patients are now added in Figure 2b.    

-              Line 367: lung injury or disfunction?

Authors’ response: Clarified.    

-              Line 392: the patients’ cohort was not well-defined and too many culture-negative cases were included to the study.          

Authors’ response: As already stated, this now added in the study limitations.    

Reviewer 3 Report

Comments and Suggestions for Authors

The authors discuss experimental data on semaphoring gene in sepsis based on prospective study SepsisFAT. The manuscript fits to the journal scope.
I have minor remarks. SepsisFAT as new term should be shown in Abstract. Might give it full in the title too.
In the Abstract:
Line 20: “SEMA -3A, - 3C” – the listing of gene name by numbers seems be not clear. Show it in full - SEMA3A, SEMA3C, and so on. Make shortcuts later in the text, if necessary.
Line 27-38: concluding phrase is week “might have an important … role”. Rephrase, write it precise.
Keyword list could include ‘immune response’ term.
Line 50: ‘classes 3-7’ – need comment on these classes, where other classes found.
Citation style could be updated – cite less references together, give more details – see [7-10], [11-13],[14-16]. It is a rather stylistic remark, but looks like bulk citation.
Line 76: SOFA – give abbreviation in full.
Similar remark for APACHE II (line 107)
HC might be redundant abbreviation for healthy control. I’d recommend avoiding it.
Line 122: ‘as suggested by the manufacturer’ – better ‘following manufacturer recommendations’ (change wording ‘suggested’ to ‘standard’, or ‘recommended’)
Line 143: ‘Most common comorbidities’ in Table 1 could be marked by font, underlining.
Line 146: ‘MetS score of 3, IQR’ – comment on MetS score, give a reference. IQR is for years in the paragraph above.

Line 153 - Table 1 has no comments after the table. Just mark some lines by font (bold, Italic), write a sentence after the table like ‘Table shows that most frequent comorbidity was Arterial hypertension, and top Infection was…”

Table 2. Baseline laboratory findings – looks redundant. It has no comments in the text. If it is not discussed then it is not needed. Add a phrase after the table 2, what one can see there.
Line 212 and below. Mark r coefficient by Italic font. Not need show exact p-value like p=0.009. Might write p<0.01. Too many numbers make it hard to read. Not need extra precision in p-values.
Line 216: ‘SIRS score’ – comment on it, give abbreviation in full.
Figure 3. Correlogram could be named as ‘heatmap’.

Figure 4. Panels for SEMA4D and SEMA7A show non significant values. Mark it properly (like p>0.1). It may confuse reader if value shown as p=0.94. Assume significant values should be marked as <0.001. not need to many digits in p value.
Line 255 – ‘8.5 day’ – should be integer number of days?
Figure 7 – panels (b) SEMA3C and (c) SEMA7A show interesting results on survival. I believe it is main result of the study. Make it as larger picture, add comments.
Line 393-394: “Our data supports the need for additional studies” – too common words. Rephrase.

Author Response

REVIEWER 3

The authors discuss experimental data on semaphoring gene in sepsis based on prospective study SepsisFAT. The manuscript fits to the journal scope.
I have minor remarks. SepsisFAT as new term should be shown in Abstract. Might give it full in the title too.

Authors’ response: We thank the reviewer for this comment and agree that term SepsisFAT (the name of the project) in title might be confusing, so we omitted it.

In the Abstract:
Line 20: “SEMA -3A, - 3C” – the listing of gene name by numbers seems be not clear. Show it in full - SEMA3A, SEMA3C, and so on. Make shortcuts later in the text, if necessary.
Authors’ response: Corrected.

Line 27-38: concluding phrase is week “might have an important … role”. Rephrase, write it precise.

Authors’ response: Corrected.

Keyword list could include ‘immune response’ term.

Authors’ response: Corrected.

Line 50: ‘classes 3-7’ – need comment on these classes, where other classes found.

Authors’ response: Corrected.

Citation style could be updated – cite less references together, give more details – see [7-10], [11-13],[14-16]. It is a rather stylistic remark, but looks like bulk citation.

Authors’ response: Corrected.

Line 76: SOFA – give abbreviation in full.Similar remark for APACHE II (line 107)

Authors’ response: Corrected.

HC might be redundant abbreviation for healthy control. I’d recommend avoiding it.

Authors’ response: Corrected.

Line 122: ‘as suggested by the manufacturer’ – better ‘following manufacturer recommendations’ (change wording ‘suggested’ to ‘standard’, or ‘recommended’)

Authors’ response: Corrected.

Line 143: ‘Most common comorbidities’ in Table 1 could be marked by font, underlining.

Authors’ response: Corrected.

Line 146: ‘MetS score of 3, IQR’ – comment on MetS score, give a reference. IQR is for years in the paragraph above.

Authors’ response: Corrected.

Line 153 - Table 1 has no comments after the table. Just mark some lines by font (bold, Italic), write a sentence after the table like ‘Table shows that most frequent comorbidity was Arterial hypertension, and top Infection was…”
Table 2. Baseline laboratory findings – looks redundant. It has no comments in the text. If it is not discussed then it is not needed. Add a phrase after the table 2, what one can see there.

Authors’ response: Brief description of results presented in Table 1 and 2 is now shown before each table. Tables and figures captions are formatted according to journal guidelines.

Line 212 and below. Mark r coefficient by Italic font. Not need show exact p-value like p=0.009. Might write p<0.01. Too many numbers make it hard to read. Not need extra precision in p-values.

Authors’ response: Corrected.

Line 216: ‘SIRS score’ – comment on it, give abbreviation in full.

Authors’ response: Corrected.

Figure 3. Correlogram could be named as ‘heatmap’.

Authors’ response: Corrected.

Figure 4. Panels for SEMA4D and SEMA7A show non significant values. Mark it properly (like p>0.1). It may confuse reader if value shown as p=0.94. Assume significant values should be marked as <0.001. not need to many digits in p value.

Authors’ response: Corrected.

Line 255 – ‘8.5 day’ – should be integer number of days?

Authors’ response: Corrected.

Figure 7 – panels (b) SEMA3C and (c) SEMA7A show interesting results on survival. I believe it is main result of the study. Make it as larger picture, add comments.

Authors’ response: We thank the reviewer for this comment. We have built new figures (Figure 7 showing ROC analysis and Figure 8 with Kaplan–Meier curves.

Line 393-394: “Our data supports the need for additional studies” – too common words. Rephrase.

Authors’ response: Rephrased.

Reviewer 4 Report

Comments and Suggestions for Authors

The paper entitled :”The role of immune semaphorins in sepsis – a prospective cohort study (SepsisFAT) “ by Branimir Gjurasin et al. deals with a clinical laboratory study on the measurement of various semaphorin molecules in the blood of septic patients.

Semaphorins concentration in the blood was measured by means of commercial ELISA kits and values obtained have been tentatively correlated with other biochemical and clinical parameters in order to validate the hypothesis that semaphorins can be useful markers in sepsis.

The management of sepsis in the clinics typically use various biochemical parameters and PCR and PCT in particular. Truth is that the management of sepsis is often very difficult and unsuccessful and there is a need for more specific and/or useful parameters.

These authors gave a try to semaphorins and by their results they were able to indicate that these molecules can be useful indeed. This is very interesting in principle, but to validate a new biochemical parameter the path is usually complicated. Large studies are needed. Therefore, we suggest they use statements very carefully.

“we have shown that patients with sepsis have different serum concentrations of SEMA3A, SEMA3C, SEMA3F, SEMA4D and SEMA7A from healthy controls and that their kinetics during sepsis correlate with disease severity and outcomes.” like many other parameters (even those cited in this study) are different in very ill patients and healthy persons. This is an obvious statement and far from a conclusion. 

The variation of semaphorins could be better correlated with sepsis if semaphorins levels were available in other conditions characterized by an inflammatory or an infectious status vs a septic status. Difference with healthy controls is present with many other parameters and this doesn’t help assess parameter’s specificity.

A correlation with other sepsis specific parameters or a pool of parameters could help. In general, specificity must be assessed.

Check the language. For instance, a phrase such as “Clinical characteristics, laboratory and demographic data were evaluated and presented deceptively”... sounds very strange. Why “deceptively”?!

Author Response

REVIEWER 4

The paper entitled :”The role of immune semaphorins in sepsis – a prospective cohort study (SepsisFAT) “ by Branimir Gjurasin et al. deals with a clinical laboratory study on the measurement of various semaphorin molecules in the blood of septic patients.

Semaphorins concentration in the blood was measured by means of commercial ELISA kits and values obtained have been tentatively correlated with other biochemical and clinical parameters in order to validate the hypothesis that semaphorins can be useful markers in sepsis.

The management of sepsis in the clinics typically use various biochemical parameters and PCR and PCT in particular. Truth is that the management of sepsis is often very difficult and unsuccessful and there is a need for more specific and/or useful parameters.

These authors gave a try to semaphorins and by their results they were able to indicate that these molecules can be useful indeed. This is very interesting in principle, but to validate a new biochemical parameter the path is usually complicated. Large studies are needed. Therefore, we suggest they use statements very carefully.

“we have shown that patients with sepsis have different serum concentrations of SEMA3A, SEMA3C, SEMA3F, SEMA4D and SEMA7A from healthy controls and that their kinetics during sepsis correlate with disease severity and outcomes.” like many other parameters (even those cited in this study) are different in very ill patients and healthy persons. This is an obvious statement and far from a conclusion. 

Authors’ response: We agree with the reviewer that our results should be viewed as preliminary and confirmed in larger cohorts. While the finding that patients with sepsis have statistically different serum semaphorin concentrations than healthy controls might seem obvious, to the best of our knowledge our study is the first exploring this question and here we present the first data in relatively large human cohort on the association of semaphorins with sepsis, which is a good starting point for further studies. Interestingly, not all semaphorins were increased, SEMA3A was significantly decreased in patients with sepsis. We are aware of the limitations of our study and have expanded “limitations section” including your remarks. In the revised version of the manuscript, we also tried to avoid strong conclusions as suggested by the reviewer.

The variation of semaphorins could be better correlated with sepsis if semaphorins levels were available in other conditions characterized by an inflammatory or an infectious status vs a septic status. Difference with healthy controls is present with many other parameters and this doesn’t help assess parameter’s specificity.

Authors’ response: We agree with the reviewers that it would be interesting to compare serum semaphorin concentrations in septic patients with other cohorts, such as SLE, RA, neoplastic diseases, metabolic syndrome, etc. Also, semaphorin concentrations could be influenced by predisposing chronic conditions in sepsis. However, this was out of the scope of this manuscript. We thank the reviewer for this comment, and we have added this consideration in the discussion.

A correlation with other sepsis specific parameters or a pool of parameters could help. In general, specificity must be assessed.

Authors’ response: We agree that large multicentric and multinational cohort of septic ICU patients is needed to perform analysis including multiple parameters of interests, and to assess the specificity. This is also now stated in our discussion.

Check the language. For instance, a phrase such as “Clinical characteristics, laboratory and demographic data were evaluated and presented deceptively”... sounds very strange. Why “deceptively”?!

Authors’ response: Thank you. We have checked the grammar and spelling.

Round 2

Reviewer 2 Report

Comments and Suggestions for Authors

The authors revised their ms responding to all questions and doubts raised by reviewer. The paper needs no further revisions. 

Reviewer 4 Report

Comments and Suggestions for Authors

The authors have included in discussion a paragraph about limitations of this study that adress our concern about the popolutaion examined. Due to the novelty of the study proposed we think that, stated that limitation the paper is acceptable.